# High-frequency Contactless Sensor for the Detection of Heparin-Induced Thrombocytopenia Antibodies via Platelet Aggregation

**DOI:** 10.3390/ijms232214395

**Published:** 2022-11-19

**Authors:** Nida Zaman Khan, Daniel Martin, Uwe Pliquett, Yahor Zaikou, Nacke Thomas, Doris Heinrich, J. Michael Köhler, Thi-Huong Nguyen

**Affiliations:** 1Institute for Bioprocessing and Analytical Measurement Techniques (iba), 37308 Heiligenstadt, Germany; 2Institute for Chemistry and Biotechnology, Faculty of Mathematics and Natural Sciences, Technische Universität Ilmenau, 98694 Ilmenau, Germany

**Keywords:** HIT, platelet aggregation/activation, KKO/RTO antibody, microwave contactless sensor, reflection coefficient, resonant frequency, conductivity

## Abstract

Heparin-induced thrombocytopenia (HIT), a severe autoimmune disorder, occurs in patients undergoing heparin therapy. The presence of platelet-activating antibodies against platelet factor 4/Heparin in the blood confirms patients suffering from HIT. The most widely used methods for HIT diagnosis are immunoassays but the results only suit to rule out HIT as the assays provide only around 50% specificity. To confirm HIT, samples with positive results in immunoassays are retested in functional assays (>98% specificity) that track platelet-activating antibodies via platelet aggregation. However, the protocols in functional assays are either time-consuming (due to the requirement of the detection of serotonin release) or require highly trained staff for the visualization of platelets. Here, we applied a cheap and easy-to-use contactless sensor, which employs high-frequency microwaves to detect the changes in the resonant frequency caused by platelet aggregation/activation. Analysis of change in conductivity and permittivity allowed us to distinguish between HIT-like (KKO) and non-HIT-like (RTO) antibodies. KKO caused a stronger reduction of conductivity of platelet samples than RTO. Our results imply that the high-frequency contactless sensor can be a promising approach for the development of a better and easier method for the detection of HIT.

## 1. Introduction

Heparin-induced thrombocytopenia (HIT) develops in up to 5% of patients undergoing unfractionated heparin therapy and occurs predominantly in females [1,2]. The etiology involves the presence of antibodies, which are produced in response to the ultra-large complexes formed between positively charged chemokine platelet factor 4 (PF4, CXCL4) and negatively charged heparin (H) [3,4,5]. HIT involves a severe loss of platelet counts which occurs in about 30–50% of associated cases with thromboembolic complications [6,7,8]. IgG isotype antibodies have Fc fragments that can bind and activate platelets via FcγRIIa receptors. Whereas the Fab parts of antibodies adhere to bound PF4/H complexes on platelets, the resulting immune PF4/H/Antibody complexes cross-link and activate platelets [9]. The PF4/H antibodies are divided into two subtypes, (i) type 1 (asymptomatic) binds to PF4/H complexes but does not activate platelets, and (ii) type 2 (symptomatic) activates platelets [10]. Additionally, some autoimmune HIT antibodies that bind to PF4 alone have been detected [11]. Asymptomatic antibodies normally do not cause HIT but they can be a possible risk for HIT. Symptomatic (HIT) antibodies cause HIT whereas autoimmune antibodies can trigger severe HIT in which platelet count reduces by around 30–50% in associated cases with thromboembolic complications. Recently, antibodies with similar characteristics as HIT antibodies have been detected in severe COVID-19-infected patients [12,13,14,15] as well as in some COVID-19 vaccinated cases [16]. The PF4/H/Antibody complexes can also bridge monocytes, neutrophils, and bind to endothelial cells [5], stimulating the response of the immune system, and causing an increased risk for procoagulant activity and life-threatening complications [6].

The presence of platelet-activating (HIT) antibodies in the sera confirms that patients suffer from HIT. Early and accurate diagnosis of HIT is of primary importance because failure to diagnose HIT antibodies can lead to catastrophic thrombosis if heparin therapy is continued. To detect these abnormal antibodies, highly sensitive and rapid antigen tests such as PF4 Enzyme-Linked Immunosorbent Assays (ELISA) are widely used in clinical laboratories because of their high sensitivity and rapid performance [17]. Immunoassays such as ELISA are widely used to detect PF4/H antibodies in clinical laboratories. However, ELISA is only suited to rule out HIT as it detects both type-1 and type-2 PF4/H antibodies as well as autoimmune HIT PF4 antibodies. Notably, it has been reported that around 50% of patients testing positive for ELISAs do not develop HIT [18]. To confirm HIT, the ELISA-positive samples must be retested in functional assays including Serotonin-release assays (SRA) or heparin-induced platelet aggregation assays (HIPA) [19,20,21]. Functional assays are reliable techniques as they provide around 98% specificity. The assays are carried out by stirring washed platelets or platelet-rich plasma together with the patient’s sera. If platelet aggregation is observed, the presence of platelet-activating (type 2 or autoimmune) antibodies in the sera is confirmed. However, the functional assays are time-consuming and require well-trained staffs as well as specialist laboratories [22,23].

To date, many technologies have been developed to improve the detection of HIT. Methods such as microscopy [24] and flow cytometry [25] are robust and powerful techniques but require complex staining procedures. Several HIT test kits such as quick test (STic Expert) [26], chemiluminescent immunoassays (AcuStar) [27], particle gel immunoassay (PaGIA) [28], chemiluminescence immunoassay (CLIA) [29], and Rapid Latex Enhanced Immunoassay (LIA) [30] are commercially available. However, they are complex, expensive, and time-consuming and were not suited for routine testing, especially, during the COVID-19 pandemic. In patients suspected of COVID-19 vaccine-induced thrombotic thrombocytopenia (VITT), abnormal antibodies (VITT antibodies) that behave similarly to HIT antibodies have been detected. Only PF4 ELISAs allow for ruling out VITT, whereas other available test kits are insensitive [31,32]. However, similar to HIT, patients must be additionally tested with functional assays to confirm if they develop VITT. Although functional assays are crucial in the confirmation of both HIT and VITT, the protocols for the recognition of positive results in these tests are to date not optimized, i.e., the platelet aggregation is normally visually inspected by highly trained staff or confirmed by a complex and lengthy process to determine the level of serotonin release [33].

In this study, we introduce a contactless microwave sensor for the detection of HIT antibodies via platelet aggregation/activation. Microwave resonators have gained increased attention for their use in biosensors as this method is cost-effective, non-invasive, contactless, capable of automation/simplification, easy to use, fast, and suitable for real-time monitoring while utilizing a disposable and sterilizable adapter to avoid sample contamination. This methodology allows for the determination of dielectric properties of the samples based on microwave measurements employing a waveguide cell filled with homogeneous media. It utilizes a microwave resonator for determining the reflection coefficient S11. Microwave sensing involves electromagnetic waves [34], undergoing reflection, refraction, and scattering. The scatter parameter S11 corresponds to the reflection coefficient if the transmission is zero can be measured between 1.8 and 2.0 GHz using a single port network analyzer. The S11 relates to the incident power (Pi) and reflected power (Pr) (Equation (1)) [35],
(1)$$S11\left(\mathrm{dB}\right)=10\log\frac{P_{r}}{P_{i}}$$

The dielectric loss is associated with the conductivity of the medium whereas a sole shift in resonance frequency indicates a permittivity change. Thus, the S11 is a frequency-dependent complex value where its amplitude and phase depend on the conductivity and permittivity of the sample. Properties of the resonator itself, and to some extent, the material inside the sample tube affect the measured S11 as well. The design was optimized for a maximal electrical field strength (E-field) at the position of the sample flow to gain the highest sensitivity. The real part permittivity ε′ of the material signifies the capacitance of the system, denoting the storage capability for the electrical energy of the sample whereas the imaginary part ε″ is related to its conductivity or the dielectric loss. Utilizing calibration with samples of known permittivity (ethanol-water mixture), the real part of the permittivity (ε′) of the material under test (MUT) can be obtained from the resonant frequency f_0_ (Equation (2)),
(2)$$\varepsilon^{\prime }=\varepsilon_{\mathrm{EtOH},\;30\%}+\left(\frac{f_{0,\;\mathrm{EtOH},30\%}-{\;f}_{0,\;\mathrm{MUT}}}{f_{0,\;\mathrm{MUT}}}\right)k_{1}$$

The k_1_ is obtained by linear fitting and k_2_ could be obtained by calibration using permittivity and conductivity of known substances such as ethanol and KCl [36]. The dielectric loss associated with the conductivity reduces the quality factor Q, which is the ratio of the resonant frequency f_0_ and the full width at half maximum (Δf_FWHM_), Q = f_0_/Δf_FWHM_. The conductivity of the MUT using the fitted parameter k_2_ could be finally calculated (Equation (3))
(3)$$\sigma^{\prime }=\left(\frac{1}{Q_{\mathrm{MUT}}}-\frac{1}{Q_{H_{2}O}}\right)k_{2}$$

We expected a microwave resonator could detect that platelet activation/aggregation induced by HIT antibodies. The hypothesis is that aggregated platelets show a lower conductivity due to their reduced mobility. A slight effect is also expected when antibodies bind to PF4. To induce platelet aggregates, we used a model monoclonal HIT-like antibody (KKO) that mimics human HIT antibodies and a non-HIT-like antibody (RTO) for comparison [37]. They both bind to PF4/Heparin complexes, but KKO causes platelet activation/aggregation whereas RTO does not [38]. Here, we report the possibility to distinguish between KKO and RTO antibodies by their specific different activity in causing platelet aggregation using the above-mentioned new microwave-based sensing method.

## 2. Results

### 2.1. Experimental Setup

The setup consists of a resonator connected to the vector network analyzer by coaxial cables and installed in an incubator to fix the temperature to 25 °C and minimize noise. Samples (MUT) were pumped inside the tube to the resonance sensor (Figure 1A) which was connected to a vector network analyzer (VNA) via the coaxial probe. KKO-induced platelet aggregation/activation is expected to be distinguishable from the RTO which does not cause platelet aggregates (Figure 1B).

The reflection coefficient S11 was recorded between 1.8 and 2.0 GHz to ensure that a large shift of resonant frequencies caused by platelet aggregation could be measured. The change in the electrical properties of the platelet sample after aggregation and activation causes a shift in the frequency (Figure 1C). Concerning the sensitivity of the sensor, it is important to concentrate on the E-field where the tubing platelet sample is situated. To identify an optimal position to place the sample tube, we simulated the distribution of the electromagnetic field using WIPL-D software utilizing the finite element method [39]. The model electromagnetic field was excited at the SubMiniature version A (SMA) connector and after being reflected by the metallic walls of the resonator, the electric component of the field was formed and distributed (green/yellow, Figure 1A). Thus, we placed our sample in the region of the maximal electric field (E-field).

### 2.2. Visualization of Platelet Aggregation by Confocal Laser Scanning Microscopy (CLSM)

It has been known that KKO causes platelet activation/aggregation in the heparin-induced platelet aggregation (HIPA) test [40]. We first confirmed that platelet aggregation is induced by KKO but not RTO. HIPA tests were performed in which antibodies were mixed with platelets and stirred for 45 min. to induce aggregation before transferring to a petri dish. Platelets were fixed and stained with mouse anti-CD42a-FITC before visualizing with a confocal laser scanning microscopy (CLSM). The platelet alone (red, Figure 2A) and RTO (Figure 2B) did not show any major aggregation, but strong aggregates were induced by KKO (arrows, Figure 2C).

### 2.3. Platelet Aggregation Causes a Change in Resonant Frequency

Platelet samples after HIPA tests were then loaded into the resonator for tracking the changes in resonant frequency caused by antibodies of different characteristics. Before starting the experiment, the calibration of the coaxial cable was carried out by standard open, short, and load calibration [41] to remove background noise caused by DC resistance from the coaxial cable. The frequency was then set in the range of 1.8 to 2 GHz with 1000 points and swept for 30 measurements with 20 s of the gap among measurements. As expected, the resonant frequency vs. S11 plot shows a shift of frequency to the right for the sample containing KKO (red) whereas RTO (blue) did not show any significant difference as compared with the platelet alone (violet) sample (Figure 3A). The negative magnitude of S11 (or f_0_) was highest for the platelet alone, followed by RTO (non-aggregated platelets), and lowest for KKO (aggregated platelets) (Figure 3A). The resonant point of spectra was different along with a slight shift of frequency, indicating that the change in conductivity and permittivity due to platelet aggregation that was induced by KKO. The S11 around f_0_ from independent platelet donors exhibited the same trend. Samples containing KKO showed the lowest negative S11 values, followed by RTO, and the highest for the platelet alone (Figure 3B).

### 2.4. Determination of Geometry Parameters of the Sensors

To understand the effect of KKO and RTO in platelet samples, we calculated conductivity and permittivity. The resonant curves were fitted to the Lorentz model, which yields the resonant frequency and the quality factor. Using linear regression overall calibration data, the geometry factors k_1_ and k_2_ were obtained. The resonator is geometry dependent, and therefore, we calculated the permittivity and conductivity through the geometry factors k_1_ and k_2_ (Equations (2) and (3)). The k_1_ and k_2_ could be obtained by calibration using the permittivity and conductivity of known substances such as ethanol and KCl [36], and thus, we performed a dilution series of ethanol (Figure 4A) up to 30% and KCl (Figure 4B) in water up to 150 mM. Equation (2) gave us a quasi-linear fit. The resonant curve was simulated with an inductor-capacitor and resistor circuit (LCR) for the determination of the geometry factors by titration curves. The k_1_ and k_2_ values were determined to be 1546 (no unit) and 1.24 × 10^6^ µS/cm, respectively.

### 2.5. Platelet Aggregation Alters Sample Conductivity

The calculated geometry parameters were used for the determination of the conductivity of platelet samples following equation (3). The results showed that platelet aggregates caused by KKO induced a lower conductivity than the non-aggregated platelets induced by RTO (Figure 5A), whereas there were no significant permittivity differences among the samples (Appendix A). To further understand the effect of KKO and RTO in the platelet samples, we calculated the variation of conductivity,
(4)$$\delta_{v}=\frac{\delta_{\left(\mathrm{KKO}\;\mathrm{or}\;\mathrm{RTO}\right)}-{\;\delta }_{\left(\mathrm{Platelet}\right)}}{\delta_{\left(\mathrm{Platelet}\right)}\;}\times 100$$

KKO-induced conductivity change (Equation (4)) is higher than that of RTO which shows a boundary at ~17% (Figure 5B). The boundary was identified by the average value obtained from the maximal and minimal δ_v_ values for RTO and KKO samples, respectively. We also calculated the ratio of conductivity change (δ_c_) to better understand the effect of KKO and RTO in the platelet samples,
(5)$$\delta_{c}=\frac{\sigma_{\left(\mathrm{KKO}\;\mathrm{or}\;\mathrm{RTO}\right)}}{\sigma_{\left(\mathrm{Platelet}\right)}}$$

This way of calculation (Equation (5)) allows for distinguishing KKO from RTO at every tested platelet donor (Figure 5C), indicating the high sensitivity of the method. Statistical comparisons show a significant change of δ_c_ between KKO and RTO, whereas no significant difference is obtained between the platelet alone and RTO (Figure 5D). The results indicate that the microwave contactless sensor allows for distinguishing HIT-like antibodies from non-HIT ones.

## 3. Discussion

The present study has demonstrated as proof of principle, the application of a contactless microwave sensor for the detection of HIT-like antibodies via conductivity change or frequency shift. To confirm HIT, the current detection procedure requires a combination of both immunoassays and functional tests. The contactless resonator microwave sensor can be a promising technique to shorten the protocol for the detection of HIT. We found a boundary of approx. 17% variation in conductivity that allows us to distinguish between a sample containing HIT-like antibodies and non-HIT antibodies. Our method shows a potentially inexpensive electronic instrumentation and label-free protocol for HIT detection via platelet aggregation in a single test.

Measurements were carried out in the frequency range of 1.8 to 2 GHz in the resonator system. Depending on materials that cause dissimilar permittivity and conductivity in platelet samples, the resonant frequency will be shifted under an applied e-field. A tube for pumping the platelet sample through is placed within the resonator, which is connected to a network analyzer to measure the reflection coefficient S11 around the resonant frequency (f_0_). The S11 depends on the permittivity, conductivity of the samples, and geometry factors of the set-up. Although permittivity influences the resonance frequency, any change in conductivity will change both the resonance frequency and the quality factor (Q) due to damping. The resonator allows for fast and low-cost detection using a portable device and the method requires only a small sample volume. Importantly, it does not cause sample contamination because it is a non-contact method. The vector network analyzer is a robust and effective instrument for the design and testing of resonator-based biosensors.

The measured resonant curve is used to determine the quality factor of the system. It is obtained by resonant frequency (f_0_) and the full width at half maximum Δf_FWHM_. Using the resonances measured in ethanol and KCl, for which permittivity and conductivity values are known, the geometry factors (k_1_ and k_2_) could be obtained by fitting their titration curves. Knowing geometry factors, permittivity, and conductivity in the platelet samples with the presence of KKO or RTO can be determined.

The sensor was applied to distinguish platelet aggregates induced by the HIT-like KKO antibodies and compared with that of non-HIT-like RTO antibodies. The ability of these antibodies in inducing platelet aggregation was confirmed by confocal microscopy. The measured resonant frequency vs. S11 plot for samples containing KKO shifted to a higher frequency compared to those containing RTO or only platelets, indicating KKO-induced changes in the permittivity and conductivity of platelet samples. Fitting resonant frequency vs. S11 plot together with using the determined geometry factors (k_1_ and k_2_), permittivity, and conductivity of platelet samples were calculated. In comparison with RTO, KKO caused a higher change in conductivity, which is likely due to the aggregation and activation of platelets. Whereas activated platelets released multiple molecules that reduce the charge mobility in the sample buffer, aggregated platelets lead to the reduction of the platelet density that causes low damping. KKO induced a strong reduction of conductivity but RTO did not. As human HIT antibodies show similar characteristics as KKO, our results suggest that HIT sera may also show an above 17% variation in conductivity. In the other words, our method can be a promising approach to confirming HIT when tracking the conductivity changes in the presence of the patient’s sera.

We did not observe a significant change in the permittivity of the platelet samples (Appendix A). It is likely that the method only allows for the detection of permittivity dissimilarity if the samples have a significantly high difference in permittivity. Among our investigated platelet samples including platelets alone and platelets with RTO or KKO, the change in permittivity is probably too small to be detected.

Though the resonator has been previously tested with inorganic materials, [28] we showed in this study an application of the sensor to the most sensitive and complicated biological samples, that is, the detection of platelet aggregation caused by HIT-like antibodies. Unlike other cell types, platelets are fragile and without a nucleus, therefore, they adhere to many non-biological surfaces and quickly activate/die. The microwave contactless sensor measures the changes in the sample medium that contains all proteins and other molecules released after platelet aggregation/activation. The experiments do not require keeping platelets in their native forms, which eliminates multiple control steps.

We presented a concept that allows portable and economic instrumentation, fast measurement, and simple operation. In comparison with current technologies such as microscopy and flow cytometry, the contactless microwave sensor does not require sample staining. Our method shows a high sensitivity for the differentiation between symptomatic HIT-like antibodies and asymptomatic non-HIT-like ones (Figure 5). With the exception of measurements with bubble effects and freshness of platelets, all tests in our experiments showed distinguishable conductivity change (δ_v_, δ_c_) of KKO from RTO. In comparison with ELISAs, our protocol is much shorter. When compared with platelet aggregation tests, the contactless microwave sensor does not require well-trained specialists or additional tests such as serotonin assays to confirm sample status caused by platelet-activating antibodies [19,20,21]. Although other available methods [18,19,20,21,22,23,24,25,26,27,28,29,30] including quick tests are of high cost or single use only, our method-based tubing system can be reused for multiple tests.

Besides having many advantages, the method still shows several technical limitations that require further development and optimization. First, we observed that the concentration of platelets and the freshness of platelets had an effect, as in some data sets, or only a slight change of conductivity was induced, or none at all. (Appendix A). To effectively induce platelet aggregation, it is important to use fresh platelets immediately after isolation from whole blood. Identification of more stable cell types for the test can be very helpful. Second, our current setup could not overcome the bubble issue as measurements in the appearance of bubbles did not allow us to distinguish KKO from RTO. Control of external factors such as tube material, the position of measurement, cable resistance, and measurement buffer can minimize bubble issues. Third, the minimization of signal-to-noise ratio as the resonator is highly sensitive to geometry and the external factors should be investigated. Dipole moment studies can further enhance the dielectric property, and thus, the sensor portability could be improved. Furthermore, advanced studies to integrate other techniques as well as automate the measurements will bring the method closer to applications at the clinics. Lastly, only HIT-like antibodies were tested in this study, and therefore, future investigation of antibodies spiked with serum or HIT patient sera will confirm the feasibility of microwave contactless sensors in the confirmation of HIT. The antibody concentration in the patient’s sera must be investigated in order to determine the detection limit. Furthermore, the effect of heparin concentration on platelet aggregation needs to be investigated as HIT antibodies enhance binding to PF4/H complexes compared to PF4 alone. Nevertheless, the microwave contactless sensor can be a potential method for diagnosing HIT.

## 4. Conclusions

A microwave resonator is presented for the detection of HIT-like KKO antibodies via changes in the characteristics of platelet samples. The HIT-like antibodies (KKO) could be well differentiated from the non-HIT antibodies (RTO) through analysis of changes in resonant frequency and conductivity. The assessment of the reflection coefficient does not require bulky and expensive equipment. The measurement yields different reflection behavior for samples with RTO or KKO added. Aggregated platelets (induced by KKO) showed a lower conductivity compared to non-aggregated platelets (caused by RTO). Evaluation of the conductivity change allows us to confirm HIT at the boundary of approximately 17% variation in conductivity. Even though human sera have not yet been tested, our study indicates that contactless microwave sensors can be a potential technology for the development of a new method for better diagnosis of HIT.

## 5. Materials and Method

### 5.1. Chemicals and Reagents

RTO and KKO from Thermo Fisher Scientific(Aachen, North Rhine-Westphalia, Germany), platelet factor 4 from Chromatec (Greifswald, Mecklenburg-Vorpommern, Germany), Phosphate Buffer Saline (PBS) pH 7.4, ultrapure water, suspension buffer, Apyrase from SIGMA (Munich, Bavaria, Germany), ACD-A (acid citrate dextrose) from Fresenius Kabi, (Bad Homburg, Hessen, Germany), washing solution (pH 6.3 composed of 137 mM NaCl, 2.7 mM KCl, 11.9 mM NaHCO_3_, 0.4 mM Na_2_HPO_4_, 2.5 U/mL Hirudin), suspension buffer, tissue culture Petri dishes, and 96 well-plates from TPP, (Trasadingen, Schaffhausen, Switzerland). All the chemicals were either prepared with the highest purity or commercially available.

### 5.2. Ethics

The use of blood obtained from healthy volunteers was approved by the ethics board of Thuringia, Germany.

### 5.3. Isolation of Platelets

Human blood from healthy volunteers (38627/2019/30-Ethics committee of Thuringia, Germany) who were drug-free within the previous 10 days were collected into a tube of ACD-A 1.5 mL BD-Vacutainer (Fresenius Kabi, Bad Homburg, Hessen, Germany). Collected blood tubes were covered with parafilm and inclined at a 45-degree angle for 15 min. After that, the blood was centrifuged at 120× *g* for 20 min to obtain platelet-rich plasma that was then collected and added a solution of 15% ACD-A along with 2.5 U/mL Apyrase, followed by centrifugation for 7 min at 650× *g*. The platelet pellets were suspended in 5 mL of washing solution and kept in incubation at 37 °C for 15 min. After centrifugation, the supernatant was discarded, and platelets were suspended with a suspension buffer. After 45 min of incubation at 37 °C, platelets were ready to use. The final concentration of platelets was 300 × 10^3^/µL was counted by blood counter pocH-100i from SYSMEX, (Möhnesee, North Rhine-Westphali, Germany).

### 5.4. Platelet Aggregation/Functional Assay

To induce platelet aggregation, the sample components consisted of fresh platelets at a final concentration of 300 × 10^3^/µL, and KKO or RTO of 10 µg/mL were added to 96-well plates. Samples were prepared in a suspension buffer. Platelet samples were stirred with two stirrer beads at 2200 rpm for 45 min at room temperature. After stirring, the samples were transferred to protein-nonbinding Eppendorf tubes and used immediately. The platelet alone without antibodies added was tested as the Control.

### 5.5. Confocal Laser Scanning Microscopy (CLMS) to Visualize Platelet Aggregation

To observe platelet aggregations, the above samples after platelet aggregation tests were transferred to Petri-dishes and kept for 15 min. Subsequently, 4% of paraformaldehyde was added to fix the platelets. After washing with PBS, anti-CD42a-FITC from Biolegend, San Diego, CA, USA) (1:1000 dilution) was added to the samples before imaging with the Zeiss LSM710 (Carl Zeiss, Gottingen, Lower Saxony, Germany) at room temperature in the dark.

### 5.6. High-Frequency Contactless Sensor Measuring Platelet Aggregation

The whole setup was placed inside an incubator set at 25 °C to keep the temperature constant. The coaxial cable was connected to the resonator via port 1 of the Vector network analyzer Master MS2028B from Anritsu, (Munich, Bavaria, Germany). A calibration of open, short, and load was carried out to match the analyzer and cable. The tube was then placed across the sensor and firmly held by two screws. The samples (200 µL) from the platelet aggregation tests were transferred to the test tube and diluted in the water of a 1:1 ratio before pumping into the resonator with a 1.0 mL/min pushing speed. Samples were kept in the incubator for 30 min to gain a constant temperature of 25 °C. Note that the waste end-tube was also closed to reduce the temperature variation and the closer was only removed after finishing the measurement to release the sample to the waste container. For all measurements, a frequency range between 1.8 and 2 GHz with 1000 points sweep was chosen. For each experiment, a set of 30 measurements were carried out with a gap of 20 s. The total time observed including incubation and measurement was 40 min for each sample. The analysis was carried out in MATLAB (The Math Works, Inc., Natick, MA, USA) and Origin Pro (version 2021. OriginLab Corporation, Northhampton, MA, USA).

### 5.7. Conductivity and Permittivity Determination

To determine the geometry factors k_1_ and k_2_, it is important to calibrate the system with known conductivity and permittivity. As any well-characterized solution could be used for this purpose, we chose KCl with dilutions up to 150 mM and ethanol titration to 30%. The whole setup of resonant frequency was fitted in the LCR circuit with the Lorentz model. Then, the values were fitted with a linear fit and the k_1_ and k_2_ were obtained. Quality factor (Q) and resonant frequency (f_0_) were obtained by fitting with the Lorentz model. The conductivity and permittivity values were calculated according to Equations (2) and (3) with the obtained Q values and f_0_ for each sample. The calculations were taken with respect to water as it was the major solvent for the titration of KCL or ethanol as well as the suspension buffer for platelets. The program was written in MATLAB (The Math Works, Inc., Natick, MA, USA) for obtaining Q, f_0_, k_1_, and k_2_. The statistical analysis was carried out in Origin Pro2021 (version 2021, OriginLab Corporation, Northhampton, MA, USA).

## Figures and Tables

**Figure 1 ijms-23-14395-f001:**
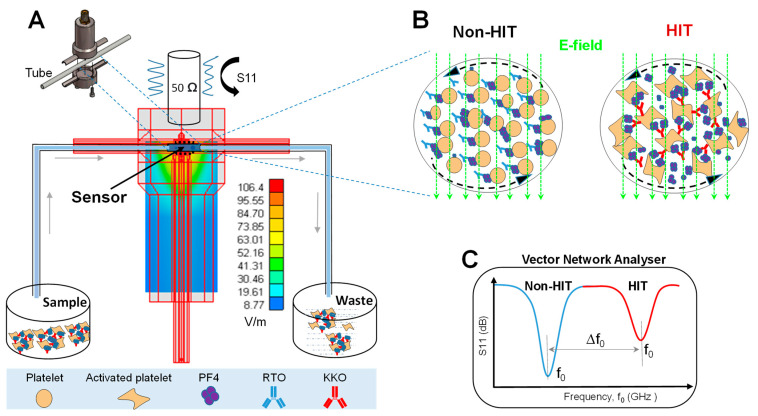
Schematic describes the setup of the contactless microwave sensor for the detection of HIT. (**A**) Simulation of electric field distribution (green/yellow) shows the maximal electromagnetic field at the site of the sample tube (top-left, enlargement). The tube was filled with platelets by pumping the samples (**left**) through the resonator into the waste container (right). (**B**) Sample components including platelets, PF4 released from platelets, and RTO or KKO under E-field are oriented according to dipole moments. (**C**) The expected resonant frequency vs. S11 plot shows a shift of frequency (Δf_0_) measured in platelet samples caused by (red) KKO (HIT) compared to (blue) RTO (non-HIT).

**Figure 2 ijms-23-14395-f002:**
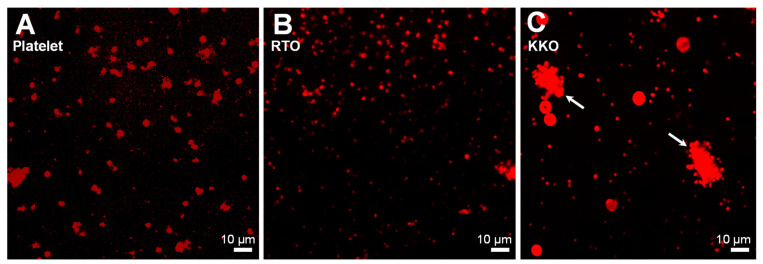
Visualization of platelet aggregation by confocal microscopy. No or very small aggregates and only minor change in platelet morphologies were seen in (**A**) platelet alone and (**B**) RTO, which can be considered as the background, but (**C**) KKO strongly caused platelet aggregates (arrows). The scale bar is the same for all images.

**Figure 3 ijms-23-14395-f003:**
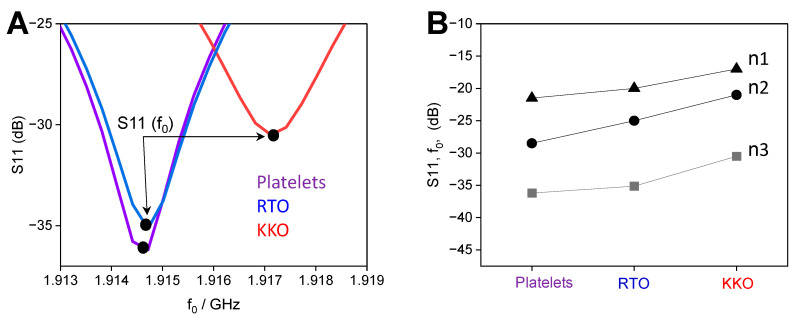
Detection of platelet aggregates with contactless microwave sensor. (**A**) Enlargements show typical−distinguished S11 vs. frequency curves for platelets (purple), RTO (blue), and KKO (red) that allow determining maximal S11 around resonant frequency f_0_, S11(f_0_), arrows. (**B**) Collection of values from different platelet donors (n = 3) shows the same trend, that is, the lowest negative S11 for KKO, followed by RTO, and highest for the platelet alone.

**Figure 4 ijms-23-14395-f004:**
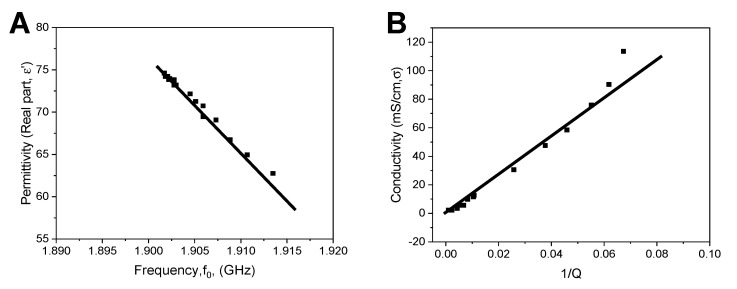
Calibration curves for determination of geometry parameters. (**A**) Ethanol and (**B**) KCl titration into water respectively allow the determination of permittivity and conductivity that enable the calculation of geometry parameters k_1_ and k_2_.

**Figure 5 ijms-23-14395-f005:**
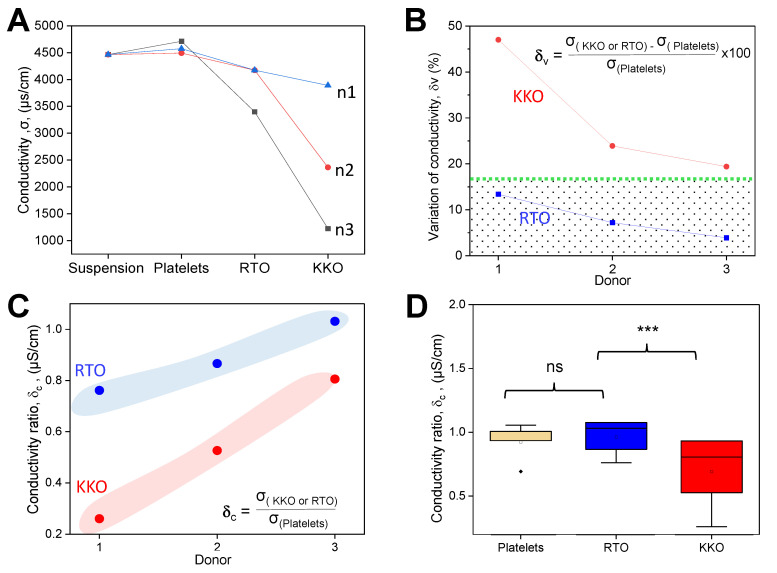
Platelet activation altered sample conductivity. (**A**) Platelet donors show the strongest reduction of conductivity caused by KKO, followed by RTO, whereas the platelet alone does not cause any change as compared with the buffer (n = 3). (**B**) Variation of conductivity (δ_v_, inset) for (red) KKO is higher than that of the (blue) RTO boundary at ~17% (green). (**C**) Change in conductivity (δ_c_, inset) from shows distinguishable σ_c_ of (red) KKO from (blue) RTO. (**D**) Statistical comparisons (T-test) show a significant change of σ_c_ between KKO and RTO, whereas no significant difference is obtained between the platelet alone and RTO (n = 5 repeats). ***: *p* < 0.001.

## Data Availability

Data is not publically available to due the institute policy but can be obtained at an reasonable request from the corresponding author.

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
