# Peer review of "High-frequency Contactless Sensor for the Detection of Heparin-Induced Thrombocytopenia Antibodies via Platelet Aggregation"

_ijms, 2022, doi:10.3390/ijms232214395_

Round 1
Reviewer 1 Report
Relevant topic dealt with well. A well designed clinical trial may not only help patients but also subjects suffering from platelet function disorders
Author Response
Response: We thank the Reviewer for the positive comments. We have significantly improved English and content in this version (please see our changes marked in red).
Reviewer 2 Report
In their manuscript with the title “High-frequency Contactless Sensor for the Detection of Heparin-Induced Thrombocythemia Antibodies via Platelet Aggregation”, the authors describe a proof-of-concept method to detect platelet aggregation in presence of HIT (Heparin Induced Thrombocythemia)-inducing antibodies. This method relies on the use of a contactless microwave sensor placed next to the sample that is subjected to microwave irradiation at very high frequencies. Presence of platelet aggregates induced by HIT-inducing antibodies is detected by a dip of the S11 parameter at higher frequencies than the dip observed with non-aggregating platelets. The authors advance that this setup may speed-up the diagnosis of HIT in the clinics.
Overall, the manuscript is very well written, and the methods are carefully laid out. I cannot make any comments on the setup or design of the detector itself as I am not an expert in microwave technology. I have some major comments about the general experimental design:
1. The authors have tested their detector using plasma from 3 healthy donors. In order to induce HIT-like aggregation, they use commercially sourced KKO (anti-PF4) antibodies. As a control they also use RTO (anti-PF4 but non-HIT-like inducing) antibodies. While this allowed the authors to verify the base functionality of their system, testing their setup with plasma from patients that have been diagnosed for HIT (through a proven methods) would be of great benefit.
2. It the same line as the comment above, comparing the results obtained to different already-established techniques of HIT detection would be a great boon to this work. It would help to solidify the point that the authors make that this method is reliable and could replace longer and more costly techniques currently used in the clinics.
3. The authors state that they found a conductivity change boundary at 12% that distinguishes between platelet-aggregated (KKO, >12% change) and non-aggregated samples (RTO, < 12% change) compared to simple platelet containing samples (Fig 4). However, they advance no details as to how they established this boundary (statistics?).
4. Like above, the whole manuscript lacks any statistical proof of significance. Can the authors show any statistical proof that the differences in conductivity and S11 parameters between KKO and RTO samples are significant?
Some minor comments:
1. Line 102: the symbol for permittivity appears as a spiral, instead of an epsilon. Same thing I presume on line 235: spiral symbol right after ‘half maximum’
2. Figure 1: a scale for the colors of the electrical field should be added in A. The schematic could also use a better legend. Not all that read the manuscript are familiar with such as setup. In B: I assume the blue squares are PF4? A legend is also needed.
3. Can the authors say how long it takes to analyze one sample? I assume 30 measurements x 20 seconds apart = 10 minutes, but this is not counting the actual time for the measurement. This is kind of important if the authors want to advance their argument that such a setup will speed the diagnosis of HIT in the clinics.
Overall, I believe the authors succeed in demonstrating that their device can detect HIT. However, more experiments (using actual HIT plasma from patients and comparing to other methods) would greatly improve the impact of this work. In its current state, the manuscript is of low impact to the field. Before publication, I would like to see some proof of statistical significance if possible.
Author Response
Dear Reviewer 2,
We thank you very much for your comments that helped us to improve the quality of our manuscript. Please find attached PDF file for our point-by-point response (red) to your comment (black).
Best wishes,
On behalf of co-authors
Dr. Thi Huong Nguyen

Reviewer 3 Report
The use of sensors for a rapid confirmation of symptomatic HIT is of high usefulness, and can be a very interesting point of care diagnostic tool for this heparin therapy complication. It can allow a rapid confirmation of platelet activation/aggregation induced by heparin dependent antibodies. However, a validation using sufficient plasma samples from patients with confirmed HIT, and documented for their reactivity in immunoassays and functional assays, is necessary for supporting this objective. This report is then very preliminary, and presents a potential tool for a rapid confirmation of symptomatic HIT diagnosis. Only KKO (a HIT-like MoAb) and RTO antibodies are used for that validation. This article is then a very preliminary feasibility report.
Although the measurement principle is correctly reported and presented, materials and methods are too poorly described for understanding how pertinent are the results obtained. The authors must indicate the protocol used in a more detailed way, and must describe their procedure with a much greater accuracy.
In addition, the scientific wording used for presenting the HIT disorder, and the involvement of heparin dependent antibodies in disease development, is approximate, and must be revised to comply with the usual standards.
Lastly, all studies show that heparin dependent antibodies activate and aggregate platelets in presence of a low concentration of heparin (in the range 0.1 to 1.0 IU/ml, and more usually at about 0.3 IU/ml), whilst in the absence of heparin or in presence of a high concentration (10 to 100 IU/ml, and usually 100 IU/ml) no platelet activation/aggregation occurs. Did the authors perform their studies in these conditions? Did they confirm their results by testing some plasmas from patients with symptomatic HIT, in the defined assay conditions?
Concerning the specific comments:
In abstract, the term of HIT antibodies is not usual, "heparin dependent antibodies" or "antibodies to HPF4" (antibodies to heparin-platelet factor 4 complexes) should be preferred. The term of "accuracy" for defining performances of immunoassays and functional assays is not appropriate, and the wording of "sensitivity" or "specificity for symptomatic HIT" fits better.
In introduction, the authors must clearly present the association of heparin dependent antibodies with disease: many antibodies are asymptomatic, whilst others are symptomatic and can produce thrombocytopenia, and in some cases thrombosis. HIT occurs only in this last case.
On line 40, note that all IgGs contain Fc residues, which can bind to platelet Fc-Gamma-RIIa platelet receptors when bound in immune complexes, and not only "some contain Fc residues".
On line 51, note that presence of asymptomatic heparin dependent antibodies do not mean HIT (but a possible risk). All text, lines 51-66 requires attention and rewriting. The same for lines 71-74.
The introduction can summarize the scientific and medical field concerned with HIT, but it must mainly focus on the reported technology itself.
The conclusion is not appropriate, as this study just demonstrates that the reported technology can differentiate between the KKO and RTO actions on platelets, but not for HIT patients (at least with the data presented). It must be revised.
Materials and methods must be much more accurate. Was heparin used for these studies? What is the volume of KKO or RTO solutions used, as only the concentration of 10 µg/ml is reported?
Author Response
Dear Reviewer 3,
We thank you very much for your comments that helped us improve our manuscript's quality. Please find the attached PDF file for our point-by-point response (red) to your comments (black).
Best wishes,
On behalf of co-authors
Dr. Thi Huong Nguyen

Round 2
Reviewer 2 Report
The authors have answered most of my concerns in their revised manuscript.
Author Response
Dear Reviewer 2,
We thank you very much for the time that you spent providing us with your helpful comments. Our manuscript has significantly improved based on your comments.
Our best wishes,
Dr. Thi Huong Nguyen on behalf of all authors
Reviewer 3 Report
Many thanks to the authors for commenting adequately the review and for all the changes introduced, which make the article better understandable, and more forcused on the potential possibilities of this technology, which is only a new "candidate diagnostic method for HIT" at this stage. In this context, the comparative KKO/RTO studies remain valuable.
I just have few comments.
In introduction, please do not use "some antibodies contain Fc residues", but "IgG isotype antibodies have Fc fragments".
The only potential use of this method for fdiagnosing HIT must be clearly stressed in the discussion, and the absence of studies with HIT patients' plasma must be highlighted. The next steps must be suggested clearly.
Author Response
Authors response to Reviewer 3 comments
Many thanks to the authors for commenting adequately the review and for all the changes introduced, which make the article better understandable, and more forcused on the potential possibilities of this technology, which is only a new "candidate diagnostic method for HIT" at this stage. In this context, the comparative KKO/RTO studies remain valuable.
I just have few comments.
- In introduction, please do not use "some antibodies contain Fc residues", but "IgG isotype antibodies have Fc fragments".
Response: We thank the Reviewer 3 for your very helpful comments that helped us to improve our manuscript.
We now corrected this statement in the introduction. The new sentence is now ‘IgG isotype antibodies have Fc fragments that can bind and activate platelets via FcγRIIa receptors.’
- The only potential use of this method for fdiagnosing HIT must be clearly stressed in the discussion, and the absence of studies with HIT patients' plasma must be highlighted. The next steps must be suggested clearly.
Response: In the original version, we have already stated in the discussion the absence of patients samples as below:
Lastly, only HIT-like antibodies were tested in this study, and therefore, future investigation with HIT patient sera will confirm the feasibility of microwave contactless sensors in the confirmation of HIT.
We now modified and added a sentence to discuss this point:
- At the end of the discussion: Nevertheless, the microwave contactless sensor can be a potential method for diagnosing
- At the end of the conclusion: ‘Even though human sera have not yet been tested, our study indicates that contactless microwave sensors can be a potential technology for the development of a new method for better diagnosis of HIT.’
Regarding the next steps, we have discussed in the last paragraph of the discussion the limitation and also our suggested solutions to improve the current status.
Besides many advantages, the method still shows several technical limitations that require further development and optimization. First, we observed that the concentration of platelets and the freshness of platelets had an effect, as in some data sets, no or only a slight change of conductivity was induced (Fig. S1B). To effectively induce platelet aggregation, it is important to use fresh platelets immediately after isolation from whole blood. Identification of more stable cell types for the test can be very helpful. Second, our current setup could not overcome the bubble issue as measurements in the appearance of bubbles did not allow to distinguish KKO from RTO. Control the external factors such as tube material, the position of measurement, cable resistance, and measurement buffer can minimize bubble issues. Third, the minimization of signal-to-noise ratio as the resonator is highly sensitive to geometry and the external factors should be investigated. Furthermore, advanced studies to integrate other techniques as well as automate the measurements will bring the method closer to applications at the clinics. Lastly, only HIT-like antibodies were tested in this study, and therefore, future investigation with HIT patient sera will confirm the feasibility of microwave contactless sensors in the confirmation of HIT.
We added now one more sentence at the last paragraph (marked in red):
- The dipole moment studies can further enhance the dielectric property, and thus, the sensor portability could be improved.
- Lastly, only HIT-like antibodies were tested in this study, and therefore, future investigation of antibodies spiked with serum or HIT patient sera will confirm the feasibility of microwave contactless sensors in the confirmation of HIT. The antibody concentration in the patient's sera must be investigated in order to determine the detection limit. Furthermore, the effect of heparin concentration on platelet aggregation needs to be investigated as HIT antibodies enhance binding to PF4/H complexes compared to PF4 alone. Nevertheless, the microwave contactless sensor can be a potential method for diagnosing
